

**Austral Summer MJO Forecast Skill in S2S Models: Decadal Shifts and Their Drivers**
Raina Roy [a,b*], Julie M. Arblaster [a,b], Matthew C. Wheeler [c], Eun-Pa Lim [c], Jadwiga. H. Richter [d]
[a]Securing Antarctica's Environmental Future, Monash University, Clayton, VIC, Australia
[b]School of Earth, Atmosphere and Environment, Monash University, Clayton, VIC, Australia.
[c]Research, Bureau of Meteorology, Melbourne, VIC, Australia.
[d]National Center for Atmospheric Research, Boulder, Colorado, USA



*To whom **correspondence** should be addressed. Email: **raina.roy@monash.edu**

**Short title: Decadal Variability in MJO Forecast Skill**

**Keywords: MJO, Prediction Skill, S2S Forecasts, QBO, IOD, ENSO, IOBM**
**ADDRESS FOR COMMUNICATIONS**


















**Highlights**


- ● The MJO exhibits greater inherent predictability in 1981-1998 compared to 1999-
- 2018, primarily due to stronger MJO amplitude (higher signal-to-noise ratio).
- ● Climate forcings (QBO, IOD, ENSO) play a primary role in modulating MJO skill in the
- models, overcoming model mean state biases.
- ● Peak skill in MJO prediction is found when easterly QBO (EQBO) winds coincide with
- negative IOD or La Niña, a synergy that weakened after 1998.




**Abstract**

The Madden–Julian Oscillation (MJO) is a key driver of global subseasonal-to-seasonal (S2S) climate variability, initiating teleconnections that affect weather patterns worldwide. Improving understanding of the factors that constrain MJO predictability is therefore critical for advancing S2S forecasting systems. Using a multi-model framework, we evaluate changes in MJO prediction skill between two periods (1981–1998 and 1999–2018) during austral summer (December–February) and examine the processes underpinning these differences. Our analysis reveals a pronounced decadal decline in MJO forecast skill, with high-skill years in 1981–1998 showing prediction lead times of around 10 days longer (based on the bivariate correlation of the Real-Time Multivariate MJO (RMM) index) than in 1999–2018, while low-skill years show little change. This asymmetric reduction coincides with stronger MJO amplitude in the earlier period, despite relatively stable model mean-state biases in tropical sea surface temperatures (SSTs) and lower-tropospheric moisture. Key findings include: (1) persistent moisture biases across both periods, yet higher skill in 1981–1998, suggesting that model errors alone cannot explain the differences; (2) a stronger Quasi-Biennial Oscillation (QBO)–MJO relationship in the first period, independent of stratospheric resolution in the models; and (3) weakened coupling between the MJO and large-scale climate modes, including the QBO, El Niño–Southern Oscillation (ENSO), and Indian Ocean Dipole (IOD), in 1999–2018, indicating reduced dynamical support for prediction. These results suggest that decadal variations in MJO skill are strongly influenced by changes in the background dynamical environment. They highlight the need for S2S systems to improve representation of tropospheric processes and stratosphere–troposphere coupling, particularly when large-scale climate forcing is weak.




## 1. Introduction

The Madden-Julian Oscillation (MJO), first identified by Madden and Julian (1971, 1972), is
the dominant mode of intraseasonal tropical climate variability. Through its planetary-scale
coupling of convection and circulation, the MJO modulates global weather systems via
teleconnections and directly influences monsoons, extreme events, and extratropical
patterns (Tseng et al., 2018; Lim et al., 2021a; Stan et al., 2022; Roy et al., 2025). These far-
reaching impacts make MJO prediction vital for subseasonal-to-seasonal (S2S) forecasting
(Zhang, 2013; Jiang et al., 2020). Despite its critical role in S2S predictability, accurately
forecasting the MJO remains a persistent challenge for state-of-the-art dynamical models
(Kim et al., 2019a). Since the landmark Intraseasonal Variability Hindcast Experiment (ISVHE)
in 2014, coordinated multi-model efforts have systematically advanced MJO prediction
capabilities through improved model physics, initialisation, and ensemble strategies (Neena
et al., 2014; Vitart, 2017; Pegion et al., 2019). These advances have extended MJO prediction
skill to approximately 25-30 days in leading systems (Kim et al., 2019a).
Despite modelling improvements, persistent deficiencies in simulating MJO propagation
across the Maritime Continent (MC) continue to limit prediction skill, particularly for MC-
terminating events compared to those propagating beyond (Abhik et al., 2023). This
persistent "MC barrier effect" (Zhang & Ling, 2017) arises primarily from model deficiencies
in representing the region's complex orography, diurnal precipitation cycles, and lower-
tropospheric moisture preconditioning (e.g., Peatman et al., 2014; Gonzalez & Jiang, 2017;
Ling et al., 2019; Savarin & Chen, 2023). These issues are compounded by systematic mean
state biases throughout the tropical Indo-Pacific, including an overly dry lower troposphere
and erroneous circulation patterns over the MC region (Kim et al., 2019a; Zavadoff et al.,
2023). Furthermore, MJO prediction skill exhibits strong sensitivity to initial conditions, with
forecast reliability depending on the event's initial amplitude and genesis location,
particularly for Indian Ocean-initiated events (Rashid et al., 2011; Lim et al., 2018; Wu et al.,
2023).

While these tropospheric factors dominate MJO predictability and representation in models
(e.g., Kim et al., 2014; Lin et al., 2024), stratospheric influences also play a critical role. In
particular, the Quasi-Biennial Oscillation (QBO) modulates MJO amplitude and propagation
through stratosphere–troposphere interactions (Son et al., 2017; Nishimoto & Yonden, 2017).
This modulation also extends MJO forecast skill, with S2S models consistently showing higher
predictability during the easterly QBO (EQBO) phases, when equatorial stratospheric winds
blow from east to west (Abhik & Hendon, 2019). The physical mechanisms underlying this
modulation involve two key processes: (1) EQBO-induced cold anomalies in the upper
troposphere-lower stratosphere (UTLS) that reduce static stability and promote deep
convection, and (2) more effective cloud-radiative feedback that amplifies and sustains MJO
circulation (Marshall et al., 2017; Hendon & Abhik, 2018; Sakaeda et al., 2020; Jin et al., 2023).
Notably, the observed QBO-MJO relationship has intensified in recent decades, likely linked
to stratospheric cooling and tropospheric warming trends (Klotzbach et al., 2019). Despite
this well-documented connection, most global forecast models struggle to replicate the QBO-
MJO relationship (e.g., Kim et al., 2019b; Martin et al., 2021). Kim et al. (2019b) demonstrated





only a weak QBO-MJO relationship reproduced by most forecast systems during the 1999-
2015 period.
While the QBO's role in MJO predictability is well established, the impacts of tropical sea
surface temperature (SST) variability remain less understood and appear to be model-
dependent. Zhou et al. (2024) recently identified enhanced MJO prediction skill during boreal
winter basin-wide warm Indian Ocean SST events, mediated through intensified convective
instability. However, their analysis was restricted to model simulations from the post-1999
period. This finding suggests a potentially essential but underexplored connection between
the Indian Ocean Basin Mode (IOBM) and MJO predictability, consistent with known
thermodynamic controls on MJO propagation.
Similarly, Liu et al. (2017) demonstrated that Indian Ocean variability (particularly positive
Indian Ocean Dipole phases) might govern the upper limit of MJO predictability in the forecast
system.    However, they cautioned about the potential model overestimation of this
relationship. Notably, both analyses were restricted to shorter periods (about ~ 20 years),
leaving open questions about the robustness of these relationships across different climate
states and longer timescales. The El Niño Southern Oscillation (ENSO)-MJO relationship
exhibits particular complexity, with studies reporting improved skill during both El Niño (via
strengthened air-sea coupling; DeMott et al., 2018; Wu et al., 2023) and La Niña conditions
(Kim et al., 2018; Mengist & Seo, 2022). This suggests that oceanic mode–MJO relationships
are non-stationary, likely reflecting their sensitivity to evolving background climate states on
decadal timescales (Zhao et al, 2016).
This study addresses these knowledge gaps through a multi-model framework examining
decadal shifts in MJO predictability during boreal winter/austral summer (December–
February (DJF))—the season of peak MJO activity and strongest coupling with major climate
modes (ENSO, QBO, IOD, IOBM). Using four S2S hindcast datasets, together with
observational verification and statistical benchmarking, we:
1.  disentangle intrinsic MJO predictability from model-specific biases;
2.  quantify how the influence of large-scale climate modes on predictability varies
between 1981–1998 and 1999–2018; and
3.  diagnose the role of evolving background states in shaping the mechanisms and
thresholds of MJO predictability.

For the rest of the paper, we will describe forecast model configurations, forecast and
verification data, and analysis methods in Section 2 and present results with discussion
in Section 3. Then, we will provide concluding remarks in Section 4.
**2. Data and Methods**
**2.1 Datasets**
This study analyses four independent subseasonal-to-seasonal (S2S) hindcast datasets to
evaluate MJO prediction skills across different modelling systems: the Predictive Ocean-
Atmosphere Model for Australia Version 2 (POAMA2, Cottrill et al. 2013), the Australian



Community Climate and Earth-System Simulator—Seasonal (ACCESS-S2, Wedd et al., 2022),
the Community Earth System Model Version 2 (CESM2, Richter et al., 2022), and the Global
Earth Observing System S2S model Version 2 (GEOS-S2S-2, Molod et al., 2020). These models
were selected from the wider S2S database based on the availability of sufficiently long
hindcasts, ensemble sizes, and initialisation frequencies to permit robust skill assessment. In
particular, POAMA2 and ACCESS-S2 provide extended archives (1981–2018), forming the core
of our decadal comparison, while CESM2 and GEOS-S2S-2, available for the later period
(1999–2018), are incorporated to broaden the analysis to more recent state-of-the-art
systems and to test the robustness of the identified mechanisms across diverse model
configurations.
Key features of each hindcast—including vertical resolutions, ensemble sizes, initialisation
frequencies, and hindcast durations—are summarised in Table 1. The study employs three
primary observational datasets for verification: (1) NCEP/DOE Reanalysis II (NCEP2) for
atmospheric variables (Kanamitsu et al., 2002), (2) NOAA AVHRR Outgoing Longwave
Radiation (OLR) as a proxy for tropical convection (Liebmann & Smith, 1996), and (3) the
NOAA OISST V2 SST dataset (Huang et al., 2021). These datasets span the 1981–2018 study
period and were regridded to a consistent 2.5° × 2.5° global grid to facilitate comparison with
model outputs. All model hindcasts (POAMA2, ACCESS-S2, CESM2, GEOS-S2S-2) are regridded
to match the observational 2.5° × 2.5° grid.

### 252 2.1.2 Climate Indices

MJO activity is quantified using the Real-Time Multivariate MJO (RMM) index (Wheeler &
Hendon, 2004), which is derived from the combined anomalies of OLR and zonal winds at 200
hPa and 850 hPa. For climate mode classification, ENSO phases are identified using the Niño
3.4 index (Trenberth, 1997), with El Niño (La Niña) defined when the 5-month running mean
of SST anomalies in the Niño 3.4 region (5°S–5°N, 170°–120°W) exceeds +0.5σ (falls below
−0.5σ). The running mean is applied continuously throughout the year, ensuring that phase
classification captures the persistence of ENSO anomalies relevant to the DJF period under
analysis. IOD events are tracked using the Dipole Mode Index (DMI) (Saji et al., 1999), which
is calculated as the difference in SST anomalies between the tropical western (50°–70°E, 10°S-
10°N) and eastern (90°–110°E, equator-10°S) Indian Ocean (i.e., the western pole minus the
eastern pole). Events are classified as positive or negative IOD based on the DJF mean of the
DMI. A positive IOD event is defined when the DJF-averaged DMI exceeds +0.5σ (relative to
the DJF climatology), and a negative IOD when it falls below −0.5σ. The Indian Ocean Basin
Mode (IOBM) index is computed as the area-weighted average of SST anomalies across the
tropical Indian Ocean (20°S–20°N, 40–110°E), following Xie et al. (2009). Warm (cold) IOBM
events are identified when the standardised index exceeds +0.5σ (falls below −0.5σ). QBO
phases are determined using 50 hPa zonal wind anomalies averaged over 5S-5N (U50) from
NCEP2, with easterly (westerly) phases defined as U50 < −0.5σ (U50 > +0.5σ) (Son et al., 2017).



| Model | Pressure levels (hPa) | Temporal Range | Ensemble Members | Initialization Frequency (Days of the Month) | References |
|---|---|---|---|---|---|
| POAMA2 (Low top model) | 17 Levels (model top at 9 hPa) | 1981-2018 | 11 | 01,06,16,26 | Cottrill et al. (2013) |
| ACCESS-S2 (high-top model) | 85 levels (model top at ~ 0.01 hPa) | 1981-2018 | 3 | 01,06,16,26 | Wedd et al. (2022) |
| CESM2 (Low top model) | 32 vertical levels (model top at 2.26 hPa) | 1999-2018 | 11 | 1/week | Richter et al. (2022) |
| GEOS-S2S-2 (high-top model) | 72 vertical levels (model top at 0.01hPa) | 1999-2018 (data unavailable for early 2017) | 4 | 1/week | Molod et al. (2020) Lim et al. (2021b) |

**Table 1.** Summary of S2S hindcast datasets analysed in this study
To evaluate MJO prediction skill, we compare dynamical model forecasts against both
observations and predictions from the Vector Autoregression (VAR) model developed by
Maharaj & Wheeler (2005). This statistical approach serves as a key benchmark for assessing
the performance of dynamical models, as established in previous studies (Rashid et al., 2011).
The VAR model predicts MJO evolution using initial values of the RMM indices (RMM1 and
RMM2) along with their lagged temporal variations, effectively functioning as an advanced
bivariate persistence forecast. Further details of the VAR model's mathematical formulation
and training procedures are provided in Marshall et al. (2016). This statistical benchmark
enables systematic evaluation of whether dynamical models outperform a simple empirical
statistical model. This study employs a period-stratified approach, analysing all observational
data, dynamical model outputs (POAMA2, ACCESS-S2, CESM2, GEOS-S2S-2), and statistical
benchmarks exclusively within two independent periods (1981–1998 and 1999–2018) to
address potential non-stationarities in MJO behaviour. The VAR model is separately calibrated
for each period through distinct regression coefficients (see supplementary text). By
maintaining identical period divisions across all components (models, observations, and VAR),
we control for background state variability and accurately quantify the evolution of skill across
different periods.



**2.2 Methods**

Our analysis employs bivariate correlation (B.Corr) between observed and forecasted MJO RMM indices (Rashid et al., 2011) to quantify MJO prediction skill during austral summer (DJF), focusing specifically on the subseasonal window (15–25-day leads) where operational forecasting transitions from weather to climate timescales (see supplementary text). This subseasonal window is used to compute an interannually varying skill index. A 15–25-day window to calculate the skill index is selected for two key reasons:

1. Subseasonal Focus: This window captures the critical transition period where deterministic weather forecasts lose skill, but MJO predictability remains viable, addressing the core S2S prediction challenge.
2. Signal clarity: In this range, interannual variations in MJO prediction skill are physically coherent and consistent across models, unlike at shorter leads (dominated by initial conditions) or longer leads (where noise overwhelms the signal). Broader windows (e.g., 10–30 days) were also tested, but these either biased the index toward initial-condition dependence or introduced substantial noise, reducing cross-model consistency.

To examine decadal changes in MJO predictability, we classified years into high- versus low-skill categories for each of ACCESS-S2, POAMA-2, CESM2, GEOS-S2S-2 and for 1981-1998 (ACCESS-S2 and POAMA2 only) and 1999-2018 using the 15–25-day bivariate correlation MJO skill index (see supplementary text for detailed methodology). We identified the top seven highest-scoring years as "Good MJO prediction years" (high-skill) and the bottom seven years as "Poor MJO prediction years" (low-skill) for each model-period combination. This comparison approach provides maximum diagnostic contrast between high- and low-skill regimes, allowing us to isolate the specific climate conditions (e.g., ENSO, QBO, IOD phases) and model characteristics that enhance or degrade MJO forecast skill. Selecting seven years per category corresponds to approximately the top and bottom 40% of cases in each period, providing a balance between statistical robustness and clear separation of skill levels.

To investigate drivers of MJO predictability, we correlate each model's MJO skill index with key climate indices: Niño 3.4 (ENSO), Zonal wind anomalies at 50 hPa (QBO), DMI (IOD), and the IOBM Index. The climate indices are derived from observed datasets. The MJO amplitude and phase are derived following Rashid et al. (2011), ensuring compatibility with established verification frameworks. We detect and track MJO events using a modified version of the Wei and Ren (2019) methodology. While their original approach focused solely on Indian Ocean-initiated events, our implementation extends coverage to all MJO phases (1-8). This adaptation provides three advantages: (1) it captures the full spectrum of observed MJO behaviour, including Pacific-originating events; (2) it eliminates geographical selection biases that could skew model verification; and (3) it increases the sample size of detectable events, enhancing statistical robustness. Steps to classify MJO events are discussed in detail in the supplementary text. We also employ linear regression between the MJO skill indices and key variables (OLR and 850 hPa specific humidity anomalies) for both observations and model forecasts to examine the background state changes associated with high MJO skill years versus low MJO skill years. Observational data are restructured to match forecast conventions, with four monthly start dates and MJO evolution time is calculated accordingly, ensuring consistent comparison with model outputs.





## 3. Results and Discussions

### 3.1 Observed MJO characteristics

Figure 1 highlights apparent epochal differences in the modulation of observed MJO amplitude by key climate modes. During the earlier period (1981–1998; Fig. 1, top row), MJO amplitude is significantly higher during (1) negative Indian Ocean Dipole (N-IOD) compared to positive IOD, and (2) easterly Quasi-Biennial Oscillation (EQBO) compared to westerly QBO, as indicated by stippling. Amplitude is also higher during cold Indian Ocean Basin Mode (C-IOBM) years, although this difference is not statistically significant. In contrast, during the later period (1999–2018; Fig. 1, bottom row), these phase-dependent relationships undergo a marked reorganisation: the EQBO–MJO linkage strengthens, while the previously significant IOD contrast weakens and falls below statistical significance, and C-IOBM differences remain non-significant.

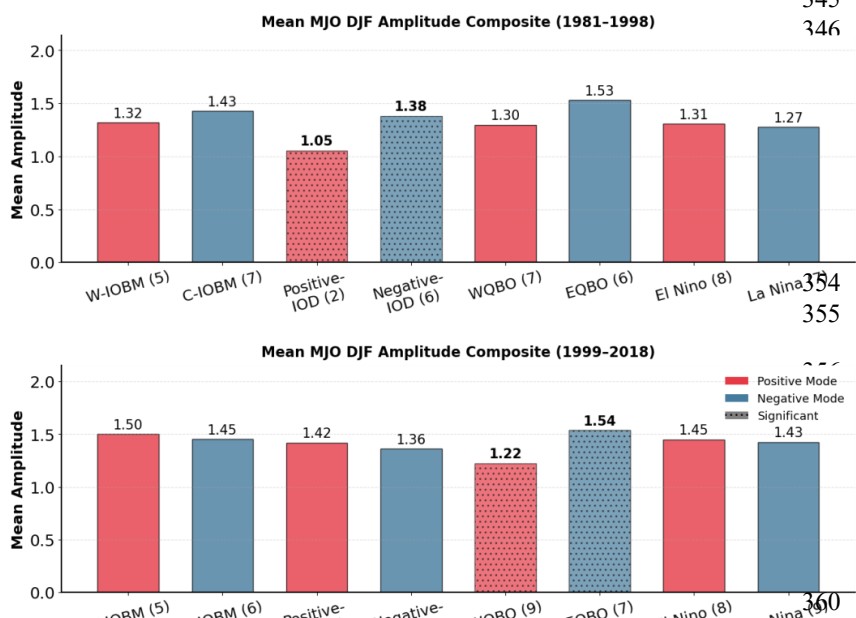

**Figure 1:** Composite mean amplitude of the MJO across different phases of climate indices— IOBM, IOD, QBO, and Niño 3.4—during austral summer. The top row displays results for the period 1981–1998, while the bottom row shows the same for 1999–2018. Numbers in parentheses denote the number of years included in each composite phase. The statistical significance of MJO amplitude differences between positive and negative phases of each index was assessed using a bootstrapping approach (10,000 iterations), where years were randomly resampled (7 per phase). Hatched bars/bold text above the bars indicate differences exceeding the 90th percentile confidence threshold.

In the earlier epoch, enhanced MJO amplitude during specific IOD and IOBM years may be partly associated with concurrent EQBO conditions. Moderate correlations between the QBO



and IOD (r = 0.32) and IOBM (r = 0.34) indices suggest some co-variability between
stratospheric and Indian Ocean conditions. Notably, IOD and IOBM are significantly correlated
(r = 0.39), indicating shared variability between these two Indian Ocean modes, while a
weaker correlation exists between QBO and ENSO (r = 0.19). Together, these associations
imply that MJO amplitude during this period was likely influenced by a combination of
stratospheric (QBO) and tropospheric (Indian and Pacific Ocean SST) factors.
While previous studies (e.g., Sun et al., 2019; Mengist & Seo, 2022; Takasuka et al., 2025) have
emphasised the role of EQBO–La Niña co-occurrence in enhancing MJO convection and
reducing the blocking effect of the Maritime Continent, it is notable that La Niña events
frequently coincide with negative IOD- and IOBM-like states (Schott et al., 2009; Cai et al.,
2011; Lim et al., 2017). The negative phase of the IOD, characterised by enhanced low-level
moisture and reduced atmospheric stability over the eastern Indian Ocean (Kug et al., 2009;
Wilson et al., 2013), provides a thermodynamic environment favourable for MJO
development, particularly in phases 1 and 2. Similarly, the cold phase of the IOBM, associated
with basin-wide SST cooling, can promote increased atmospheric instability and moisture
convergence across the Indian Ocean sector, particularly in phases 3 and 4. These conditions
likely acted in concert with EQBO to strengthen MJO amplitude during the 1981–1998 period,
highlighting a synergistic interaction between stratospheric and tropospheric drivers.
In contrast, the 1999–2018 period exhibited intensified EQBO–MJO coupling (Klotzbach et al.,
2019) but a breakdown of tropospheric linkages, with QBO–IOD/IOBM correlations
weakening to r = 0.15–0.13 and QBO–ENSO becoming slightly anti-correlated (r = −0.18). This
breakdown reflects a fundamental shift in the background state, where the loss of combined
stratospheric-tropospheric forcing—exacerbated by a weakened N-IOD–La Niña relationship
post-1999 (Zu et al., 2024)—diminished MJO amplitude modulation.
To further characterise MJO variability, we examined relationships between interannual MJO
event properties (mean DJF duration and total yearly event count for DJF) and climate mode
indices.  The QBO exerted the most substantial and most persistent influence, with MJO event
duration showing robust negative correlations (1981–1998: r = −0.67; 1999–2018: r = −0.50).
Composite analysis (Figure S1) illustrates the frequency distribution of MJO phases, showing
that in EQBO years, particularly during 1981–1998, the MJO spends more days in phases 3–6.
Although the figure does not directly plot event duration, this higher phase occupancy reflects
longer-lived events, consistent with the negative correlation between QBO and mean DJF
event duration (r = −0.67). In contrast, MJO event count exhibited weaker associations with
QBO (r = −0.28 to −0.20).
The tropospheric modes exhibited temporally varying relationships with MJO frequency: (1)
the DMI correlated negatively with event count in 1981–1998 (r = −0.48, reflecting enhanced
frequency during N-IOD years), but this relationship reversed sign and weakened post-1998;
(2) the IOBM index developed a positive correlation (r = 0.38) in the later period (more events
during W-IOBM years); and (3) ENSO (Niño 3.4) showed strong positive correlations in the
first period (r = 0.46, linking El Niño (La Niña) to increased (decreased) MJO activity) that
weakened substantially thereafter. This weakening of tropospheric mode relationships in the
second period may stem from reduced co-occurrence of favourable QBO and tropospheric
climate mode phases. Notably, none of the tropospheric indices (IOD, IOBM, ENSO)
significantly correlated with MJO duration, underscoring that while they modulate initiation





frequency, event longevity is governed primarily by stratospheric (QBO) processes. Figure S1
composites illustrate these dynamics: EQBO years in 1981–1998 featured both greater total
MJO days and more substantial phase-specific enhancement (phases 3–6), whereas
tropospheric influences (e.g., El Niño/N-IOD/W-IOBM linkages to MJO frequency) weakened
or reversed in the later period. Collectively, these results demonstrate a stark contrast:
tropospheric mode relationships with MJO evolved markedly between periods, while the
QBO's stratospheric influence remained robust, highlighting its dominant role in MJO
prediction.

**3.2 MJO skill indices in the models**

As found in previous studies, all dynamical models exhibit appreciable skill for the 15–25 day
lead time during most years, with bivariate correlations generally above ~0.5, indicating
meaningful predictability of the MJO at subseasonal timescales. Figure 2 reveals systematic
differences in MJO forecast skill between models and across the two periods. Dynamical
models show strong inter-model agreement in skill indices during both 1981-1998 (Fig. 2A)
and 1999-2018 (Fig. 2B), while the statistical VAR model (yellow line) consistently
underperforms - a pattern confirmed also by the mean skill comparison with lead time (Fig.
2C, yellow line). POAMA2 emerges as the highest-skill dynamical model (green solid line; ~26-
day skill) (Fig. 2C), potentially attributable to its enhanced MJO amplitude relative to
observations (compare green and black solid lines; Fig. 2D). It is also noteworthy that the MJO
amplitude is substantially underestimated in ACCESS-S2 (blue lines), particularly during the
second period, compared to the other models and observations. This amplitude deficiency,
however, does not translate to proportionally reduced forecast skill; the model maintains skill
levels comparable to those of other dynamical models for this period. This apparent
discrepancy suggests that while accurate amplitude representation may contribute to
forecast skill (as seen in POAMA2's strong first-period performance), other factors may play
compensatory roles in maintaining usable skill despite amplitude biases in ACCESS-S2.
A comparison of good MJO prediction years reveals substantial differences between periods
(Fig. 2E, compare the solid and dashed lines). ACCESS-S2 and POAMA2 demonstrate
approximately 10 days greater forecast skill during 1981–1998 compared to 1999–2018, with
this enhancement directly attributable to stronger MJO amplitudes in the earlier period (Fig.
2F, compare solid and dashed lines). The statistical VAR model shows a similar, though
statistically insignificant ($p > 0.05$), first-period skill advantage. In contrast, the skill difference
between the first and second periods disappears in years with poor MJO predictions (Fig. 2G,
compare solid and dashed lines), where all models exhibit comparable performance across
periods. Importantly, there is strong consistency across models in the classification of good-
and poor-prediction years, with most years falling into the same category across systems,
reinforcing the robustness of the inter-model signal. These results suggest that recent
changes in tropical climate dynamics—such as changes in the background state—have
disproportionately affected MJO prediction skill in good MJO prediction years, while leaving
poor-prediction years relatively unchanged.

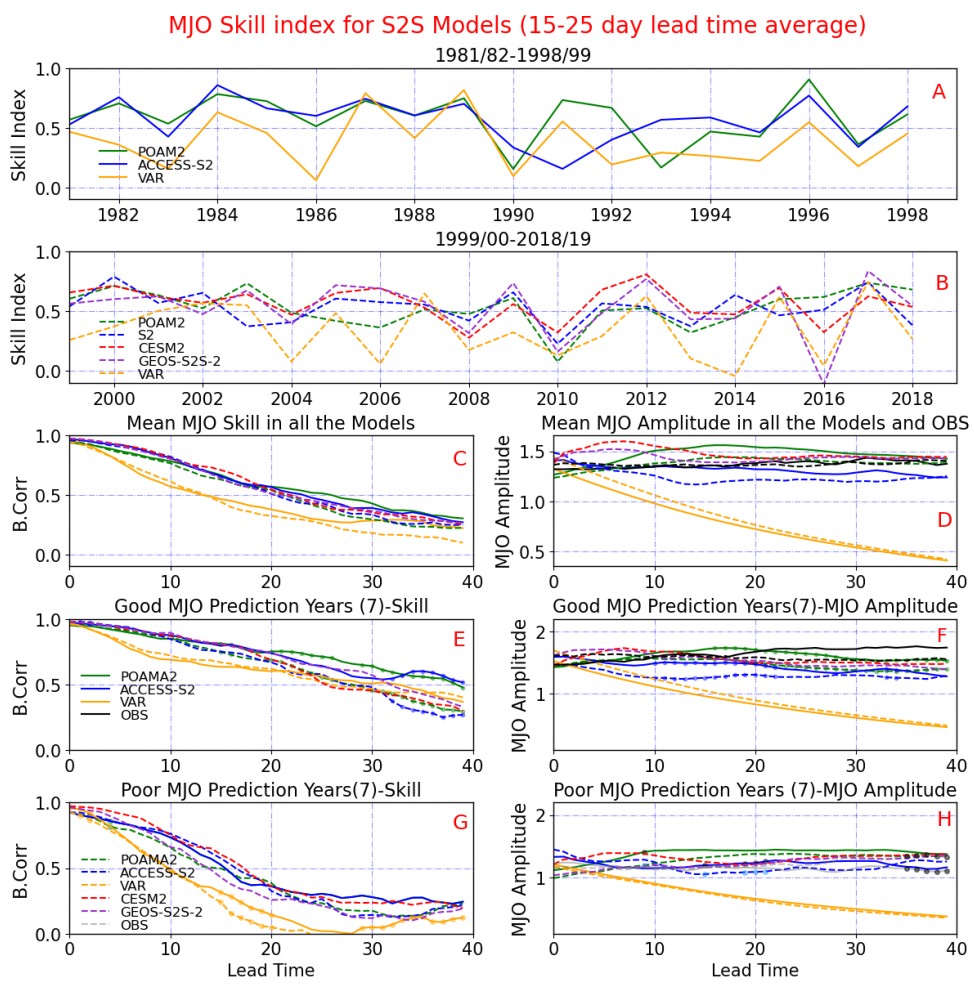

**Figure 2:** Comparison of MJO forecast skill metrics between 1981–1998 (solid lines) and 1999–2018 (dashed lines). (A, B) Skill indices at 15–25-day lead times across models (VAR, POAMA2, ACCESS-S2 in 1981–1998; CESM2 and GEOS-S2S-2 added in 1999–2018). (C, D) Composite mean skill and amplitude versus lead time. (E, F) Results for the seven highest-skill years (good MJO prediction); (G, H) the seven lowest-skill years (poor prediction years). Dots indicate statistically significant differences (p < 0.05) between periods for models present in both eras (ACCESS-S2, POAMA2, VAR). The comparison of good and poor MJO prediction years for the observed MJO amplitude is obtained using the multi-model mean skill index of the dynamical models for each period.

**3.3 Mean State Biases in the Models**

Figure 3 illustrates model mean state-specific humidity at 850 hPa and SST biases in the MJO forecast models across both study periods. All models exhibit an El Niño-like warming pattern in the eastern tropical Pacific, with POAMA2 and CESM2 showing the most substantial biases (exceeding +1.5°C) and ACCESS-S2 showing the weakest bias (< +0.8°C). The second period



(1999-2018) exhibits notable Indian Ocean warming in CESM2 and GEOS-S2S-2, contrasting
with the first period (1981-1998), during which POAMA2 and ACCESS-S2 display more
localised eastern Indian Ocean warming, accompanied by a characteristic negative IOD-like
pattern. The identified SST biases likely affect MJO skill by modifying the background state
through which the MJO propagates. The El Niño-like eastern Pacific warming may weaken the
Walker circulation, and this weakening can, in turn, promote stronger eastward propagation
of the MJO (Wang & Li, 2021). In the first period, negative IOD-like patterns in
POAMA2/ACCESS-S2 could promote more realistic MJO initiation in phases 3 and 4.

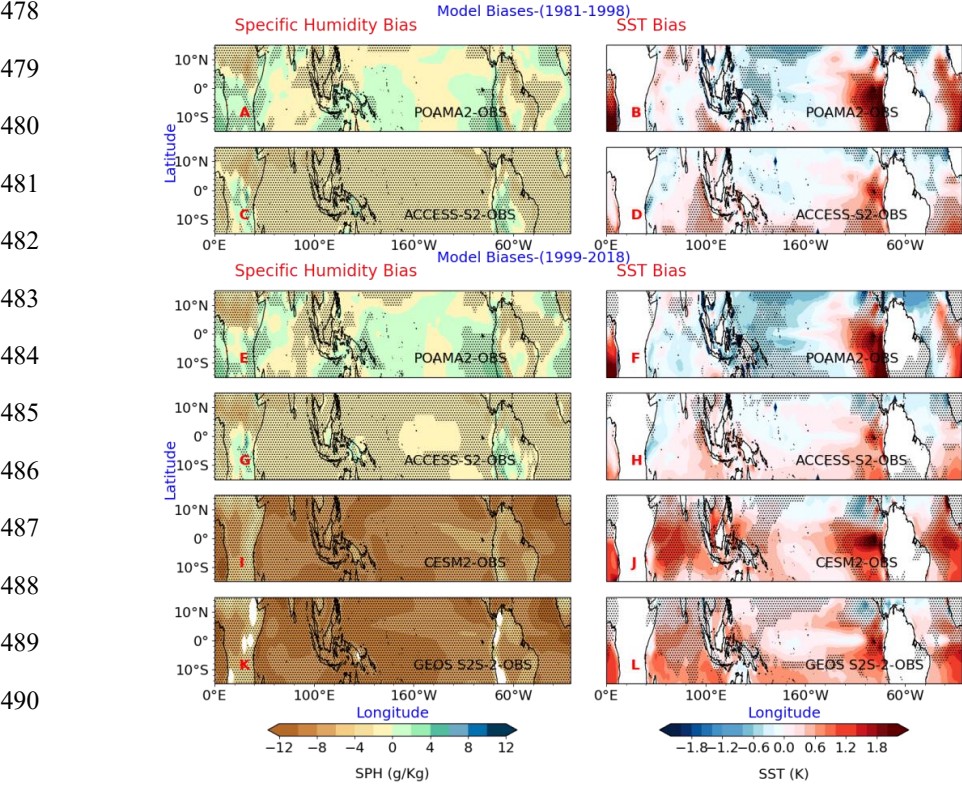

**Figure 3:** Mean state biases in MJO forecast models for 40-day lead time from the initialisation
dates, showing differences between modelled and observed climatologies for (left) specific
humidity at 850 hPa and (right) SST. Top panels (A-D) display biases for 1981-1998 from the
model ensemble (POAMA2, ACCESS-S2), while bottom panels (E-L) show 1999-2018 results,
including models (CESM2, GEOS-S2S-2). Positive values indicate that the model overestimates
the value of the variable relative to the observations. The stippling in the figure suggests
significant biases estimated using the bootstrapping method.

Inter-model and period comparisons reveal no relationship between dry biases and MJO
forecast skill. While POAMA2 and ACCESS-S2 maintain consistently weaker dry biases than
CESM2 and GEOS-S2S-2, their skill characteristics show period-dependent behaviour: both
models achieve superior performance during high-skill years in the first period (1981–1998;
Fig. 2E) but experience notable skill reduction in the second period (1999–2018) without



corresponding increases in moisture bias. Crucially, mean skill levels remain comparable
across all models despite their divergent dry bias magnitudes (e.g., CESM2/GEOS-S2S-2 vs.
POAMA2). This apparent disconnect suggests that moisture biases alone are insufficient to
explain variations in MJO forecast skill. Supporting this, Figure 3 shows positive SST biases
over the Maritime Continent and eastern Indian Ocean in some models that do not translate
into corresponding positive moisture anomalies at 850 hPa. This mismatch points to a possible
decoupling between SST anomalies and moisture–convection feedback, potentially linked to
weak surface wind anomalies or limitations in convection parameterisation. Thus, while dry
biases do not directly correlate with skill differences, improving convection schemes and their
coupling with large-scale dynamics could still be beneficial for enhancing MJO skill (Zhu &
Hendon, 2015), particularly during periods of weak external forcing. We investigate the
influence of large-scale dynamics in the following section.
**3.4 Large-scale dynamics influencing MJO skill in the models**
Figure 4 elucidates the large-scale atmospheric controls on MJO forecast skill by analysing
regression patterns between model and observed state variables and the MJO skill index
computed for individual models for respective periods. 30-day lead OLR and 850 hPa specific
humidity anomalies are regressed onto the MJO skill index (see Section 3.2). This approach
identifies characteristic patterns associated with high-skill/low-skill MJO prediction years
while enabling direct model-observation comparisons.
During the first period (1981–1998; Fig. 4, top panels), POAMA2, ACCESS-S2, and VAR
simulate a negative IOD-like pattern in the Indian Ocean, marked by lower-tropospheric
moistening in the eastern Indian Ocean (EIO) and drying in the western Indian Ocean (WIO),
alongside collocated OLR anomalies showing enhanced EIO convection and WIO suppression.
This dipole structure suggests that higher MJO skill coincides with a background state
replicating an IOD-driven moisture-convection feedback. In the Pacific, models exhibit weak
La Niña-like signatures in humidity and OLR, although these are statistically insignificant (p>
0.10). ACCESS-S2 shows the strongest La Niña signal, while VAR captures only marginal Pacific
anomalies. Collectively, these patterns indicate that first-period MJO skill is optimised when
models simulate (1) a negative IOD-like regime in the Indian Ocean, enhancing moisture
convergence, and (2) weak La Niña-like Pacific Ocean conditions, supporting a stronger
Walker circulation.







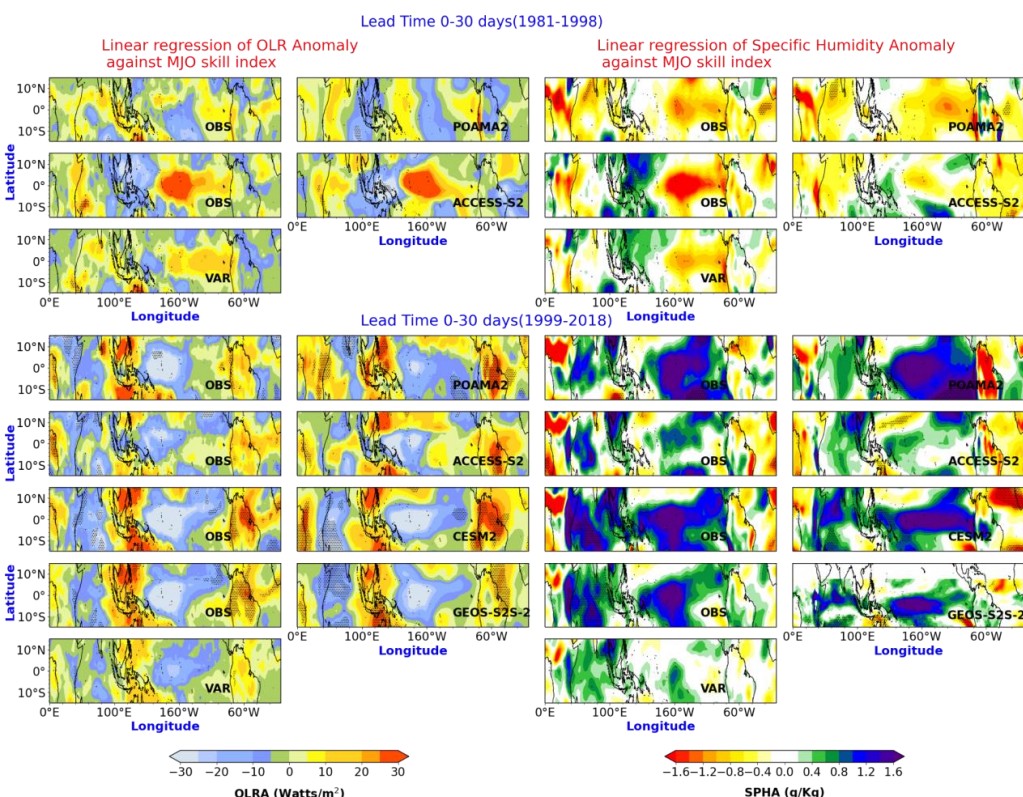

**Figure 4:** Linear regression of OLR anomalies and 850 hPa specific humidity anomalies onto MJO skill indices computed for individual models and periods. The top panels display results for the period from 1981 to 1998, while the bottom panels show results for the period from 1999 to 2018. Stippling marks regions where regression coefficients are statistically significant (p < 0.10).

During the second period (1999–2018; Fig. 4, bottom panels), the models simulate distinct shifts in the relationships with the background states. POAMA2, CESM2, and GEOS-S2S-2 exhibit strong moistening in the central Pacific, resembling a central Pacific El Niño pattern (Ashok et al., 2007), with collocated OLR anomalies showing enhanced convection; however, these linkages remain statistically insignificant (p> 0.10). In contrast, ACCESS-S2 and VAR lack a clear El Niño-like signature. The Indian Ocean displays divergent model behaviours: most models (excluding POAMA2, ACCESS-S2, and VAR) align with a warm IOBM regime, marked by basin-wide moistening and convection, while POAMA2, ACCESS-S2, and VAR instead reflect a positive IOD-like dipole pattern. Notably, all models systematically underestimate the observed magnitudes of moistening in the Indian Ocean across both periods, suggesting a pervasive bias in representing moisture convergence dynamics in the Indian Ocean region. These results highlight how epochal changes in tropical climate modes differentially influence model skill, with central Pacific El Niño, warm-IOBM and positive IOD-like regimes emerging as competing controls on MJO predictability in the most recent period.



Our analysis reveals fundamental differences in how tropospheric background states
modulate MJO predictability across epochs. Between 1981 and 1998, both dynamical and
statistical models demonstrated higher skill under consistent background conditions—a cold
IOD-like pattern in the Indian Ocean, coupled with weak La Niña-like Pacific anomalies—
suggesting robust tropospheric control of MJO predictability. Post-1998, this coherence
breaks down: models diverge in their responses to background states, with POAMA2, CESM2,
and GEOS-S2S-2 tracking central Pacific (CP) El Niño-like conditions, while ACCESS-S2 and VAR
exhibit either positive IOD linkages or weak tropical Pacific connections. This reduced
tropospheric influence likely stems from the following:
1.  Increased sensitivity to Indian Ocean warming in recent years in some models
(Dalpadado et al., 2021);
2.  ENSO diversity, particularly increased CP El Niño events (Freund et al., 2019), which
alter MJO propagation pathways (Chen et al., 2015) and
3.  Stratospheric dominance, as the QBO's role in MJO skill, intensifies due to the
absence of coherent tropospheric drivers.

The declining inter-model agreement further underscores that contemporary MJO prediction
skill may depend more on interannual variability in stratospheric processes (e.g., QBO) than
on the tropospheric background state. This has important implications for model
development.
To elucidate the evolving relationship between large-scale drivers and MJO predictability, we
computed correlations between the multi-model mean skill index and the climate mode
indices (averaged across dynamical models) during austral summer (December-February) for
each period separately. This analysis reveals how the MJO-climate mode linkage has changed
between 1981-1998 and 1999-2018, identifying which climate condition became more or less
influential on MJO forecast skill over time. During 1981–1998, the multi-model mean skill
index exhibited the strongest correlation with the QBO index (r = −0.41, p < 0.01),
demonstrating enhanced predictability during EQBO years, with EQBO conditions supporting
extended predictability windows of 25-35 days. This was complemented by weaker but
consistent relationships with negative phases of the ocean-atmosphere coupled modes: IOD
(r = -0.36, p < 0.01), ENSO (r = -0.30), and IOBM (r = -0.21), indicating that La Niña and negative
IOD/IOBM conditions further enhanced predictability when coincident with EQBO. These
correlation patterns strongly support our earlier findings that optimal MJO predictability
occurred during periods when negative IOD/IOBM/ENSO phases coincided with EQBO
conditions. This enhanced predictability primarily results from strengthened MJO convection
during phases 3-6, when the MJO's convective envelope interacts most strongly with the
Indian Ocean-western Pacific warm pool. The combined effects of (1) EQBO-induced
stratospheric wind modulation and (2) warmer SSTs under negative IOD/IOBM/ENSO phases
create favourable conditions for enhanced lower-tropospheric moisture convergence and
deep convection. Both observational composites (Figure S1) and model simulations confirm
this amplification mechanism, demonstrating more vigorous MJO activity in phases 3-6 during
these co-occurring climate mode conditions (not shown). Post-1998, while the QBO influence
strengthened further (r = -0.46, p < 0.01), its practical impact diminished as predictability
windows shortened to 21-26 days during EQBO years. This paradox stems from a breakdown
in the previously synergistic QBO-ocean-atmosphere coupling, as evidenced by reversed
correlations with IOD (r = +0.34), ENSO (r = +0.31), and IOBM (r = +0.17) and weakened phase





alignments. This reversal occurred alongside a breakdown in favourable phase alignments
(QBO-tropospheric mode correlations < 0.15), leaving no coherent multiscale forcing regime.

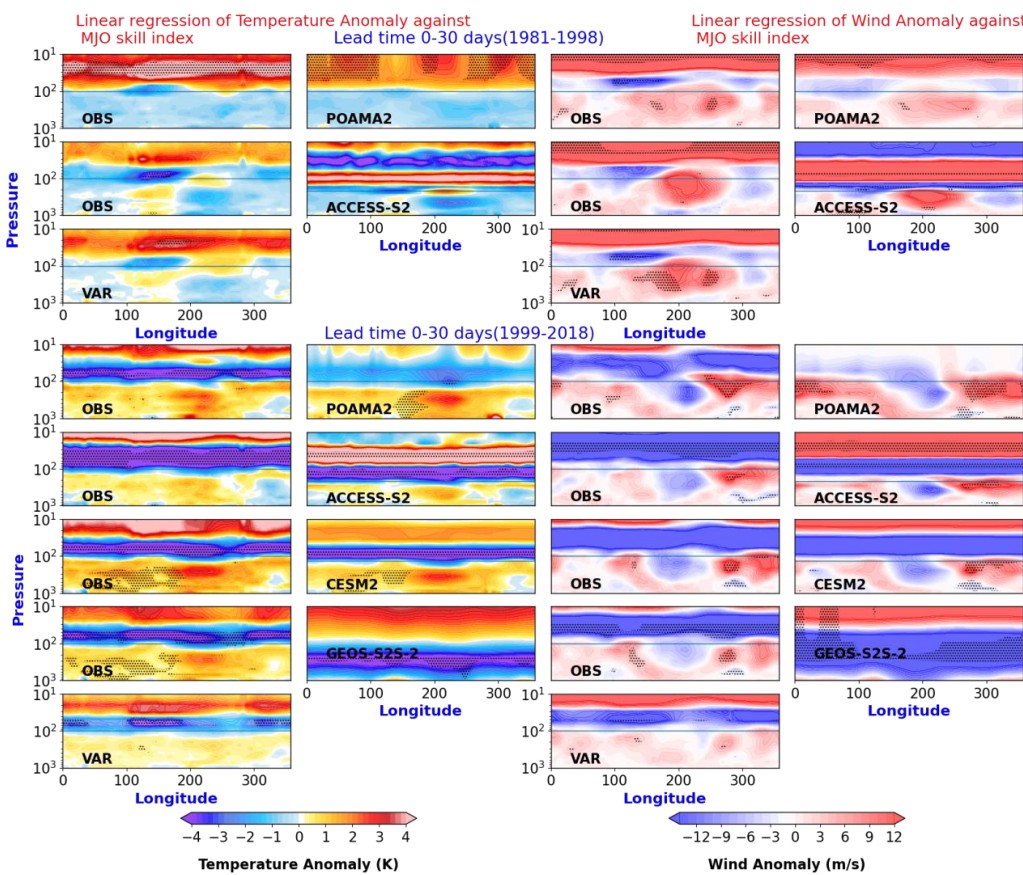

**Figure 5:** Linear regressions of observed and modelled vertical profiles of (a) temperature and
(b) zonal wind anomalies onto each model's MJO skill index for both periods (1981-1998 and
1999-2018). The top panels present regression patterns for the earlier period (1981-1998),
while the bottom panels show results for the later period (1999-2018). Stippling indicates
regions where regression coefficients are statistically significant at the 90% confidence level
($p < 0.10$).
Figure 5 validates the identified MJO skill-background state relationships through vertical
structure analysis, examining temperature and zonal wind anomaly profiles via linear
regression with model skill indices. During 1981–1998 (top panel), all models simulate EQBO-
like zonal wind patterns (easterly anomalies <-5 m/s in the lower stratosphere) which results
in tropopause-level (100-200 hPa) temperature instability ($\Delta T$ ~-2.2–-1.2 K), though with
notable inter-model differences: ACCESS-S2 better resolves these stratospheric signatures
compared to POAMA2, which exhibits weaker vertical coherence due to its poor stratospheric
representation. These patterns intensify post-1998, with wind anomalies strengthening and
temperature instability increasing by ~1–2 K, indicating enhanced QBO-MJO coupling. This





vertical structure analysis aligns with our skill-index correlations, where the QBO relationship
strengthened (r = −0.41 to −0.46). The consistency between these independent diagnostics
(regressions and correlations) demonstrates that MJO predictability has shifted from being
governed by coordinated tropospheric-stratospheric drivers in the first period to increasingly
stratosphere-dominated controls in the second period, with model skill strongly dependent
on faithful representation of QBO-related vertical coupling processes.

**3.5 Relationship Between MJO Event Characteristics in the Models and the Climate Indices**

Figure 6 displays the correlation coefficients between interannual MJO characteristics in the
models (mean MJO event duration and total MJO event count for every DJF) and observed
climate indices. In the first period, ACCESS-S2 (compared to POAMA2 and VAR) shows a
significant negative correlation (−0.48) between the QBO index and MJO event count, a
stronger relationship than observed.

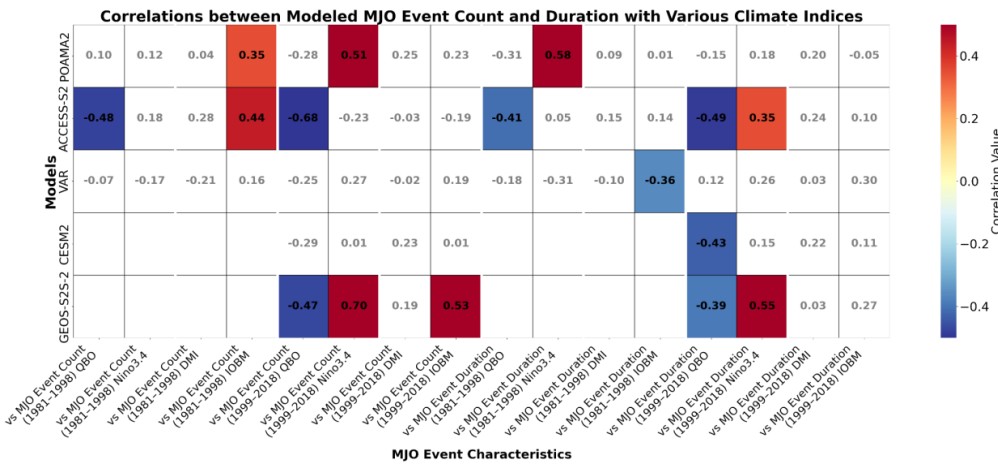

**Figure 6:** Correlation between MJO event characteristics (total MJO event count and mean
annual event duration) against various climate indices (QBO, Niño3.4, DMI and IOBM for
different periods.
In the second period, the relationship between MJO event counts and the QBO index
strengthens in ACCESS-S2 (−0.68) and is also evident in GEOS-S2S-2 (−0.47). This suggests a
higher MJO event count during EQBO years. This is unlike the case in the observations.
Additionally, moderate negative correlations (−0.39 to −0.49) emerge between MJO event
durations and QBO indices in CESM2, ACCESS-S2, and GEOS-S2S-2. However, these model-
simulated relationships underestimate the observed relationships between the QBO and MJO
duration linkages, except for ACCESS-S2. Because the QBO is skilfully predicted by the models
(Table 2), these model-dependent relationships between the MJO characteristics and the
QBO appear to originate not from stratospheric forecast errors, but from misrepresented
tropospheric responses.
During the second period, both POAMA2 and GEOS-S2S-2 exhibit significant positive
correlations between the Niño3.4 index and MJO event count (0.51 and 0.70, respectively),



in contrast to observations, which show no such relationship. This discrepancy is also evident
in the first period, as neither model captures the observed moderate positive MJO event
count-ENSO connection (see section 3.1). Additionally, these models overestimate the
relationship between MJO event duration and the ENSO index, suggesting that simulated MJO
events persist longer during El Niño years or that the models exhibit a bias toward El Niño-
like conditions. The El Niño-like warming bias in the Pacific Ocean region observed in Section
3.3 may be indicative of the same phenomenon. Klingaman and DeMott (2020) reported a
similar result, finding increased MJO activity during El Niño years in CMIP-class models
(specifically SPCAM3 and SPCCSM3), a pattern largely attributed to the East Pacific warming
bias present in these climate models.
The relationship between MJO event count and the DMI index during the first period differs
substantially from observations in all models except VAR. While observations indicate an
increase in MJO event frequency during negative IOD years, models generally fail to capture
this association. In the second period, some models produce weak correlations between MJO
event frequency and positive DMI, unlike in the observations. Notably, observations reveal
no significant relationship between MJO event duration and IOD phases in either period.
However, during the second period, most models generate weak (albeit non-significant)
positive correlations between MJO duration and DMI, suggesting a potential positive IOD-like
bias, with VAR and GEOS-S2S-2 as notable exceptions. The moderate skill in representing DMI
variability across models (Table 2) further implies that the fidelity of MJO-IOD linkages in
models may be tied to their ability to simulate IOD behaviour realistically.
A moderate positive correlation exists between MJO event count and the IOBM index during
the first period, suggesting that warm IOBM phases (W-IOBM years) are associated with
enhanced MJO activity in model simulations. However, observational records show no
significant MJO-IOBM relationship, implying this connection may be a modelling artifact
rather than a genuine climate feature. While Table 2 indicates that most models reproduce
the IOBM index with reasonable accuracy, this apparent skill in representing Indian Ocean
variability does not extend to the MJO-IOBM linkage, suggesting potential oversimplifications
or errors in the modelled physical mechanisms. During the second period, models generally
underestimate MJO event frequency during warm IOBM phases, with GEOS-S2S-2 being the
notable exception - its overestimation of this relationship may stem from the warm bias in its
Indian Ocean simulation (Section 3.3). Additionally, CESM2 performs distinctly worse than
other models in simulating IOBM-like conditions, as evidenced by its lower correlation (0.69)
between observed and modelled IOBM indices. The Indian Ocean warming bias identified in
CESM2 in Section 3.3 also supports the lower correlation.









| Model | QBO | DMI | IOBM | Niño 3.4 |
|---|---|---|---|---|
| ACCESS-S2 1981-1998 | 0.96 | 0.73 | 0.95 | 0.80 |
| POAMA2 1981-1998 | 0.84 | 0.80 | 0.93 | 0.81 |
| ACCESS-S2 1999-2019 | 0.94 | 0.53 | 0.83 | 0.83 |
| POAMA2 1999-2019 | 0.86 | 0.67 | 0.83 | 0.93 |
| CESM2 | 0.97 | 0.67 | 0.69 | 0.98 |
| GEOS-S2S-2 | 0.98 | 0.55 | 0.70 | 0.82 |

**Table 2:** Correlations between climate mode indices computed from the observed datasets
and model datasets for the two periods
**4. Conclusion**
Accurate prediction of the MJO is crucial for advancing S2S forecasting capabilities, given its
global impact on tropical convection and extratropical teleconnections (Vitart & Robertson,
2018). Our analysis of four S2S models (POAMA2, ACCESS-S2, GEOS-S2S-2, CESM2) and a
statistical benchmark (VAR) during austral summer (December–February) across two periods
(1981-1998 (POAMA2, ACCESS-S2)) vs. 1999-2018 (POAMA2, ACCESS-S2, GEOS-S2S-2,
CESM2)) highlights three key findings.
First, we assessed how the MJO interacts with various climate modes by conducting a
composite analysis of observed MJO amplitude during different phases of the climate modes.
In the earlier period, the IOD exhibited a pronounced influence on MJO amplitude, with the
negative phase associated with more vigorous MJO activity. This enhancement was further
supported by increased MJO amplitude during EQBO years and cold phases of the IOBM.
However, in the later period, the relationship between the negative phase of the IOD and
MJO amplitude disappears, leaving the MJO modulation primarily associated with the EQBO.
Additionally, during the first period, EQBO–MJO events tended to coincide with favourable
tropospheric states, such as negative IOD, cold IOBM, or La Niña-like conditions, suggesting a
coherent multi-mode influence. By contrast, in the second period, the connection between
stratospheric and tropospheric climate modes is diminished, and QBO–MJO events occur
largely independently of favourable tropospheric forcings. The annual count of MJO events in
observations shows a moderate correlation with coupled ocean–atmosphere modes, while
the mean yearly event duration correlates directly with EQBO conditions.
In the second part of our study, we analysed the relationship between MJO prediction skill
and large-scale climate modes to verify the characteristics identified in observational records.
The mean prediction skill for the MJO in ACCESS-S2 and POAMA2 showed minimal difference



between the two periods examined. This difference between the periods, however, is not
observed in the years with poor MJO predictions. The elevated skill observed in the high-skill
years of the first (earlier) period was primarily linked to higher MJO amplitude. These years
showed a clear association with EQBO-like signatures in the stratosphere, a negative IOD–like
pattern or cold IOBM-like pattern in the Indian Ocean sector, and, to a lesser extent, La Niña–
like conditions in the Pacific. This phase alignment among the stratospheric process,
processes in the Indian and the Pacific Oceans provided a favourable multiscale environment
that enhanced both MJO amplitude and model predictability. By contrast, in the second (later)
period, high-skill MJO years exhibited distinctly different background conditions. For these
years, Indian Ocean SST patterns resembled warm IOD/IOBM–like modes, and the Pacific
showed El Niño–like features—conditions less conducive to amplified MJO activity. As a
result, the previously robust synergistic connection between different climate modes
appeared to break down, resulting in diminished MJO predictability even in otherwise "high-
skill" years. Notably, EQBO remained the only significant and persistent modulator of MJO
skill in the second period, but its overall impact was weaker without concurrent favourable
patterns in other climate modes.
We speculate that in the later period, the weaker QBO–MJO relationship in model forecasts—
despite the QBO being reasonably well represented in both high-top (ACCESS-S2, GEOS-S2S-
2) and low-top (POAMA2) models, and consistent with the earlier period—likely reflects
changes in the tropospheric background state. In the second period, the models struggle to
capture the stratosphere–troposphere coupling, particularly in the absence of strong external
forcing, as was more commonly observed during the first period. Additionally, key processes
in the troposphere are often poorly captured by the models. For example, models analysed
in this study fail to accurately replicate the observed relationship between MJO event
frequency and duration and the ENSO index. This limitation suggests an underlying Pacific
bias, with many models skewing toward El Niño–like conditions, potentially linked to a known
warm bias in the eastern Pacific. Moreover, models (notably POAMA2, CESM2, and GEOS-
S2S-2) exhibit a tendency toward the warm phases of the IOD and IOBM, in contrast to
observed variability. These systematic biases in representing coupled ocean–atmosphere
background states may limit the models' ability to simulate realistic MJO behaviour.
Therefore, future modelling efforts would benefit from improving the representation of
tropospheric circulation patterns and reducing SST-related biases, particularly in the Indo-
Pacific region, alongside enhanced stratospheric resolution. Such improvements may provide
a pathway to improving MJO prediction skill, especially during periods lacking strong
multiscale external forcing. A key limitation of this study is the relatively short hindcast record
available, particularly for some models, which constrains the ability to robustly assess longer-
term variability and low-frequency influences. Extending forecast datasets further back in
time would strengthen the ability to disentangle model biases from genuine dynamical shifts
and provide a more comprehensive understanding of decadal changes in MJO predictability.
**Code and data availability**
The model data analysed during this study are available from the corresponding author upon
reasonable request. All publicly available observational and reanalysis datasets are provided
by the NOAA Physical Sciences Laboratory (PSL) and the Australian Bureau of Meteorology
(BOM) as follows:



- The NCEP-DOE Reanalysis 2 data:
  https://psl.noaa.gov/data/gridded/data.ncep.reanalysis2.html
- The High-Resolution OISST Version 2 data:
  https://psl.noaa.gov/data/gridded/data.noaa.oisst.v2.highres.html
- The Real-time Multivariate MJO (RMM) index (Wheeler and Hendon, 2004):
  http://www.bom.gov.au/climate/mjo/graphics/rmm.74toRealtime.txt
  The analysis codes and scripts used in this study are available from the
  corresponding author upon reasonable request.

**Supplement link**

**Author contributions**

R.R. conducted the formal analysis and led the original manuscript preparation. J.M., E.L., M.M., and J.R. provided supervision, critical guidance, and feedback. All authors contributed to the study's design, interpretation of results, and manuscript review.

**Competing interests**

The contact author has declared that none of the authors has any competing interests.

**Acknowledgements**

The numerical analysis was conducted using Python (version 3.9). We acknowledge the National Computational Infrastructure (NCI), supported by the Australian Government, for providing computational resources and services. We thank the NOAA Physical Sciences Laboratory for producing and making the NCEP-DOE Reanalysis II dataset publicly available. We are also grateful to Drs Zoe Gillett and Hongyan Zhu at the Bureau of Meteorology for their valuable feedback.

**Financial support**

This work was supported by Securing Antarctica's Environmental Future, funded by the Australian Research Council (ARC) Special Research Initiative in Excellence in Antarctic Science Grant SR200100005. E.L. received partial support from the Victorian Water and Climate Initiative (VicWaCI) phase 2 and the National Environmental Science Program (NESP) phase 2.



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
