# Peer review of "Austral Summer MJO Forecast Skill in S2S Models: Decadal Shifts and Their Drivers"

_EGUsphere, 2025_

## Author Comment (AC1)

RC2: 'Comment on egusphere-2025-4453', Anonymous Referee #2, 01 Nov 2025  reply

This study is evaluating the relationship between MJO and modes of climate variability in the tropics such as ENSO, IOD, IOBM, and stratospheric QBO during two periods: 1981-1998 and 1999-2018. The analysis indicates different relationships between the MJO and the other modes of variability from one period to another. Assuming that climate modes provide a source of predictability for the MJO, the second objective of the study is to test if models show a change in the forecast skill of the MJO for the two periods. While the first part is robust, the approach chosen for the second part has limitations because the POAMA2 model has a poor representation of stratospheric dynamics and the CESM2 and GEOS-S2S-2 models do not have data for the first period. There are other concerns that can be addressed and they are listed below.

We thank Reviewer 2 for their detailed feedback on our study. We acknowledge the critical limitations regarding model configuration and data availability and address each concern below. Please see the attached file for our responses to all comments.

**POAMA2 Stratospheric Dynamics:**
We recognise the limited representation of the stratosphere in POAMA2 (model top at 9–10 hPa), particularly when assessing the QBO–MJO relationship. However, we utilise this constraint diagnostically (also see Abhik & Hendon (2019) and Marshall & Hendon (2017)). A key point is that POAMA2 is an initialised forecast system: while it does not have sufficient vertical resolution to realistically simulate QBO dynamics in free-running mode, the QBO state is well-initialised at the forecast start, allowing QBO and its influence on the troposphere to be skilfully predicted (e.g., Table 2). Furthermore, the upper troposphere—where QBO–MJO interaction is thought to occur—has considerably better vertical resolution in POAMA2 than the stratosphere.

Importantly, POAMA2 nonetheless tracks observed decadal skill shifts despite its poor stratospheric resolution, highlighting the likely role of the tropospheric background state and large-scale forcing in modulating MJO predictability, often overcoming model deficiencies. In the revised manuscript, we have explicitly discussed these limitations (Section 3.4) and clarified that POAMA2's inclusion is motivated by experimental design and interdecadal continuity, and it's relatively poor stratospheric resolution has limited impact on the results. We have also:

- Given greater emphasis to high-top models (CESM2 and GEOS-S2S-2) in our physical interpretations
- Softened claims relating to QBO–MJO mechanisms in POAMA2
- Explicitly noted that POAMA2 skill likely reflects statistical or coherent tropospheric response rather than dynamically realistic stratospheric pathways

**Data Availability (CESM2 and GEOS-S2S-2):**

We confirm that CESM2 and GEOS-S2S-2 data were only available for 1999–2018. To ensure robust interdecadal comparison, we relied primarily on models spanning both periods (ACCESS-S2 and POAMA2) and incorporated CESM2 and GEOS-S2S-2 for the second period analysis to broaden the assessment to additional modern, state-of-the-art systems. These constraints were necessary to conduct a multi-model decadal comparison using currently available hindcast data. Our methodology is designed to diagnose non-stationarity in MJO prediction skill and MJO–climate mode relationships within this framework.

We will now address the other specific concerns listed below.

> L236: The three models used in the study (ACCESS-S2, CESM2 and GEOS-S2S-2) are not part of what is known in the community as the S2S data base: https://apps.ecmwf.int/datasets/data/s2s/levtype=sfc/type=cf/

We apologise for the incorrect statement regarding the S2S database. We have removed references to ACCESS-S2, CESM2, and GEOS-S2S-2 being part of the official S2S database, as they do not meet the formal classification criteria.

> Section 3.1: Please provide a table showing which years have been used for each of the phases of the climate modes shown in Fig. 1. The table can be in the supplement file.

We have included a supplementary table listing the specific years used for each climate mode phase (ENSO, IOD, IOBM, QBO) for the two periods.

> L238: Please discuss the source of initial conditions used for POAMA2 and ACCESS-S2. If they use the same initial conditions the difference in skill will be solely due to models' differences. If there was any change in the DA system used to generate the initial conditions between the two periods, that should also be discussed.

This is a crucial point regarding model independence. We have added clarification that POAMA2 and ACCESS-S2, while both are the Bureau of Meteorology system's, use different initial conditions and data assimilation (DA) systems. POAMA2 employs its own internal DA system, whereas ACCESS-S2 uses a more modern, separate system. Consequently, differences in skill arise from a combination of model physics/dynamics and initialisation differences. Critically, there was no change in the DA system within each model between the two periods (1981–1998 and 1999–2018), ensuring a consistent framework for interdecadal comparison.

> L389-394 and L419-422: Coincidently, the two periods considered in the study correspond to two phases of the Pacific Decadal Oscillation (PDO). 1981-1998 is mostly dominated by positive values of the PDO index whereas the 1999-2018 is dominated by negative values of the PDO index. This is also another factor affecting the mean state and should be mentioned when describing the shift in the background state.

This is an insightful observation and we should have clarified this in the introduction. We have now added a discussion acknowledging that the two study periods approximately correspond to the Pacific Decadal Oscillation (PDO) phase shift: 1981–1998 was dominated by positive PDO values, while 1999–2018 experienced predominantly negative PDO values. We have incorporated this as another large-scale factor contributing to changing the background dynamical environment and non-stationarity in MJO predictability.

> L396: Please explain how 'the mean DJF duration and total yearly event count for DJF' are calculated.

We apologise for not including a statement on the mean DJF duration and total yearly event count for DJF. We have now updated the MJO event estimation section in the Supplementary material. An event is defined as an occurrence of MJO amplitude >1 for at least 7 consecutive days and progression to at least 2 subsequent phases (adapted from Wei & Ren (2019)). Total yearly event count is the total number of MJO events for a given DJF, and the mean duration is the average duration for these MJO events. We have also updated the text around Line 321 to point to this new section.

> L399-400: Figure S1 shows the phases grouped as 4, 5, 6, 7 and 8, 1, 2, 3. One cannot see that 'the MJO spends more days in phases 3–6.' If this is the message that the figure is intended to convey, then the grouping of phases should be 3,4,5,6 and 7,8,1,2.

We apologies for this typographical error which was also noted by Reviewer 1. We have corrected the legend for Figure S1 to reflect the actual phase groupings used in the plot: Phases 3, 4, 5, 6 and Phases 7, 8, 1, 2, ensuring the figure clearly conveys that the MJO spends more days in phases 3–6.

> L405-407: If the negative correlation is explained by the enhanced frequency of N-IOD years the reversal of sign means an increase frequency of positive IOD years? Weakening means a lower value of r?

The interpretation is correct. We have clarified the text to explicitly address the shift: The negative correlation in 1981–1998 (r = −0.48) reflects an increase in MJO event frequency during Negative Indian Ocean Dipole (N-IOD) years. The subsequent sign reversal to a positive but weak correlation in 1999–2018 (r = 0.29) indicates a shift toward slightly higher MJO event frequency during Positive IOD years in the later period, contrasting the substantial N-IOD enhancement observed earlier. Weakening is precisely defined by correlation values moving closer to zero.

> L405-422, L579-590: I suggest summarizing all correlation coefficient values into a table.

We agree that this information is best presented in a centralised, easily digestible format, as also suggested by Reviewer 1. We have created a comprehensive supplementary table that

summarises all the linear correlation coefficients discussed in **Sections 3.1 and 3.4** (MJO event characteristics vs. climate indices, and multi-model skill index vs. climate indices) for both periods.

Figure 2: Panel B 'S2' should be ACCESS-S2

We appreciate the attention to detail. We have corrected the legend in Figure 2B to display **ACCESS-S2** instead of 'S2'

L512-514: These results should be connected to the findings of Jiang et al. (2015, https://doi.org/10.1002/2014JD022375). They also show that feedbacks between moist convection and circulation are critical for simulation of the MJO.

We agree that drawing this connection strengthens the discussion on model biases. We have added a reference and discussion to Jiang et al. (2015) to support the point that poor representation of convection schemes and their coupling to large-scale dynamics (highlighted by the SST-moisture decoupling in Figure 3) is a recognised limitation in MJO simulation.

Please explain the interpretation of regression analysis. The idea of identifying patterns associated with high/low MJO skill depends on how the low skill is defined. For example, if the correlation coefficient has a large negative value, the skill is low, but the regression coefficient will have a large value. Second, what is the reason for regressing observations onto the model skill? And lastly, the regression coefficients in Fig. 4 show very limited statistical significance, which raise the question of how robust this analysis is.

We appreciate the request for clarification. We have expanded the explanation in Section 3.4:

**Linear Regression Method:** Regressing observed fields (e.g., OLR) onto the multi-model MJO skill index identifies large-scale patterns associated with years of high model MJO skill. Positive/negative regression coefficients signify that positive/negative anomalies of the state variable tend to occur during years when models show high MJO forecast skill. This links model performance directly to real-world large-scale forcing. Regressing **observed** fields onto the **model's MJO skill index** determines whether the years a model finds easy to predict (high skill) correspond to a specific, observed background climate state. This links model performance directly to real-world large-scale forcing.

**Significance Level:** We note that the significance threshold is set at $p < 0.10$, a standard for decadal variability studies due to limited sample sizes. While individual regions show limited stippling, the discussion emphasises consistent dipole structures in the Indian Ocean (Figure 4, Period 1) and overall pattern shifts, suggesting robust structural changes across epochs despite limited local significance.

The usage of POAMA2 model for the evaluation of MJO-QBO relationship raises some questions about this model ability to resolve the stratosphere more than what is acknowledged in the study. The model top is located at 10 hPa, meaning that the model does not have a full stratosphere. Compared to the QBO lifecycle, these forecasts are relatively short, and the model might be tuned to have a 'good QBO' but miss the QBO dynamics.

We fully acknowledge and agree with the reviewer that POAMA2's relatively low model top (≈9–10 hPa) is a limitation for faithfully simulating the QBO. Models with such a low upper boundary cannot fully represent the vertical structure, wave–mean flow interaction, or downward influence of the QBO          in the same way as high-top models.

However, we note that, as these are initialised experiments, the QBO state is well represented at the beginning of the forecast, thus allowing the QBO's impact on the MJO to persist. Furthermore, the vertical resolution of POAMA 2 around the upper troposphere, where the QBO-MJO interaction is thought to occur, is much better than its resolution in the stratosphere. Therefore, there is still some utility in using POAMA2 to study the impact of the QBO on the MJO.

In the revised manuscript, we now explicitly discuss this in Section 3.4 and the Conclusions.

Importantly, POAMA2 is **not used here as a benchmark for realistic stratospheric representation**, but rather for **consistency in the decadal comparison**. This continuity enables a clean assessment of temporal changes in forecast skill under a constant model framework.

To ensure that physical interpretations of the QBO–MJO coupling are not disproportionately influenced by a low-top model, we have:

· Given greater emphasis in the discussion on the **high-top models** (ACCESS-S2, CESM2 and GEOS-S2S-2),

· **Softened claims** relating to the QBO–MJO mechanism in POAMA2,

· Explicitly noted that any POAMA2 "skill" is likely a **statistical persistence or coherent tropospheric response**, rather than a dynamically realistic stratospheric pathway.

These changes clarify that POAMA2's inclusion is motivated by experimental design and interdecadal continuity, not by its capacity to resolve the full stratospheric dynamics of the QBO.

Fig. 6 Caption: Please explain why some boxes are filled with color. On the x-axis please use font of different colors for denoting the two periods and draw a thick vertical line

between the left side of the plot with event count and the right side of the plot with event duration.

We thank the reviewer for this helpful suggestion. These changes have been fully implemented:
- Coloured boxes: The figure caption now explicitly states that filled/coloured boxes indicate statistically significant differences between periods at the 90% confidence level (p < 0.10), while unfilled boxes denote non-significant values.
- Visual Improvements:
    - Used different font colours on the x-axis to distinguish Period 1 (1981–1998) and Period 2 (1999–2018)
    - Added a thick vertical dividing line separating event count panels (left) from event duration panels (right)

These modifications significantly improve figure readability and clarify the intended comparisons between periods and MJO characteristics.

---

## Author Comment (AC2)

RC1: The article evaluates the MJO predictive skill between two periods (1981-1998 and 1999-2018) using several S2S forecasting systems. The authors found that the MJO predictive skill was smaller in the latest period, particularly during high skill years. They relate this change of predictive skill to changes in the background dynamical environment.

This article addresses the important topic of the interdecadal variability of MJO predictability and predictive skill. A better knowledge of this variability might help identify model current limitations in the prediction of the Madden Julian Oscillation. However, the presentation of this article needs to be improved. The text is not always clear. Some results seem contradictory. Therefore, I recommend major revisions.

We sincerely thank Reviewer #1 for the thorough and constructive review and for the detailed, actionable comments. Please find our point-by-point responses below.

Major comments:
1. A major limitation of this study is the too short periods considered. 20 years is likely not to be enough to investigate the link between MJO and ENSO, QBO or IOD. This is mentioned as a caveat at the end of the article, but should be further discussed. A possible explanation why the 2 periods display such different relationships between background state and MJO characteristics might be that 20 years is not enough. For instance, according to Figure 1, the number of years with positive IOD in the period 1981-1998 is only 2! It is not guaranteed that such small sample is representative of the general population. If it is not, then the bootstrap resampling method is likely to be overconfident. Figure 1 should show error bars computed over each population. I am expecting these error bars to be huge, at least for some of the climate indices. Sub-sampling between high score and low scores makes the sample even smaller. Bootstrap resampling might not be accurate over such a tiny sample. It would be interesting to compare some of the results (e.g. amplitude composites of Figure 1) with the results obtained over the 40-year period.

**1. Comment on Sample Size and Robustness**

We agree with the reviewer that sample size is a significant limitation in quantifying climate-mode modulation of the MJO, and that certain phases—particularly positive IOD events in 1981–1998—are sparsely sampled. This limitation is now more explicitly acknowledged and discussed in the revised manuscript (Section 3.1 and Conclusion).

Also, we have added error bars (standard error) in Figure 1 for all composites to clearly illustrate the uncertainty associated with small phase populations, particularly in the first period.

We have clarified the aim and scope of the paper. The purpose of dividing the record into two 20-year epochs was not to estimate a long-term climatology, but rather to examine the **non-stationarity of MJO characteristics and forecast skill on interdecadal timescales**. Our analysis was motivated by the observed decline in MJO predictive skill after the late 1990s, and the hypothesis that this change may be linked to shifts in the background thermodynamic and dynamical environment.

Importantly, since the submission of this manuscript, recent work by *Kim et al. (2025)* has independently demonstrated that **asymmetric tropical ocean warming in recent decades has altered the mean state and regional propagation of the MJO**, reinforcing the idea that MJO behaviour is not constant across epochs. Although this study did not inform our original analysis design, its conclusions are entirely consistent with and lend independent support to our interpretation that the two periods represent **dynamically distinct regimes**.

In summary, to directly address the reviewer's concerns regarding statistical robustness, we have implemented the following revisions:

1. **Figure 1 now includes error bars** (standard error) for all composites, clearly illustrating the uncertainty associated with small phase populations, particularly in the first period.

2. We now explicitly state that **bootstrap-based confidence levels should be interpreted with caution** for sparsely populated categories (e.g., positive IOD in 1981–1998).

3. Following the reviewer's suggestion, we have added **supplementary figures showing the same composite analyses over the full 40-year period (1981–2018)**.

   o As expected, these show smoother and more statistically stable signals.

4. The revised manuscript clarifies that the purpose of the two-epoch comparison is to diagnose **decadal structural changes** in the coupled system.

These additions ensure transparency regarding sample-size limitations, while strengthening confidence that the reported interdecadal differences in MJO behaviour and forecast skill reflect **fundamental, physically driven changes in the background state** rather than solely sampling uncertainty.

1. This article contains several contradictions, missing information.

Other comments:

1. Line 254: anomalies relative to what climate? Is it the same climate for Period 1 and 2 (e.g. 40-year climate) or is the climate of period 1 (2) used to compute

anomalies in period 1 (2)? If that's the case, do you remove the scoring year in the climate calculation?

We apologise for the lack of clarity. In this study, anomalies were calculated relative to the climatology of each respective epoch (1981–1998 and 1999–2018), rather than against a single 40-year baseline. This approach was adopted to isolate interannual variability within each dynamically distinct period and to avoid conflating signals from potentially non-stationary background states.

Revision: We have updated the text to explicitly state: "Anomalies are computed with respect to the climatology of the corresponding period (1981–1998 and 1999–2018)."  The scoring year was not excluded from the climatological calculation, and it will be stated in the method section.

2. Line 284: What period was used to train the VAR? Is the training period independent from the verification periods (1981-1998 or 1999-2018)?

We thank the reviewer for requesting clarification on the training methodology for the Vector Autoregressive (VAR) model used to identify the lead–lag structure between the MJO and climate indices.

In the revised Section 2.2, we will clarify that the VAR model was trained separately on each full epoch: **1981–1998 for Period 1 and 1999–2018 for Period 2**. The purpose of the VAR analysis was not to produce independent year-by-year forecasts, but rather to **diagnose the dominant linear relationships and interaction timescales** characterising the MJO–climate mode coupling within each decadal regime.

Because our objective was to examine **structural differences in the background dynamical environment** between the two periods, we consider it appropriate to use the complete 20-year block for parameter estimation. The VAR model is therefore used here strictly as a **diagnostic tool to characterise non-stationarity**, rather than as a predictive model.

We will ensure that this methodological intent and the non-forecasting nature of the VAR analysis are stated explicitly in the revised manuscript.

3. Abstract, line 119: "A stronger QBO-MJO relationship in the first period.: contradicts Figure 1, which shows a statistically significant and larger QBO-MJO amplitude relationship in the second period.

We acknowledge the confusion caused by imprecise wording. The statement incorrectly conflated two distinct relationships:

·  The observed modulation of MJO amplitude by the QBO
·  The relationship between the QBO phase and the MJO forecast skill in the models

While Figure 1 indeed shows that the observed MJO amplitude during the easterly QBO phase is stronger and statistically significant in the later period (1999–2018) , the amplitudes between the two periods are very close (1.53 in the first period and 1.54 in the second period). Hence, we have revised the text to focus only on the influence of the QBO phase on the forecast skill.

Our analysis of the model MJO forecast skill in relation to the QBO (shown in Figure 5 and associated correlation analysis, not Figure 1) indicates that the models exhibited greater influence from the QBO on MJO prediction skill in the earlier period (1981–1998), particularly when reinforced by skill-favourable ENSO and IOD conditions. In the latter period, although the observed QBO–MJO coupling is stronger, the multi-mode synergistic influence on skill weakens, as reflected in deteriorating forecast skill.
Revision: All text has been revised to reflect this point and to ensure consistency with the figures.

4. Lines 337-338: "2) easterly quasi-Biennial Oscillation (EQBO)". This statement contradicts Figure 1, which shows statistical significance for QBO only in period 2 (1999-2018). Not in period 1 (1981-1998).

Thanks for pointing out this inconsistency. The text has been corrected to align with Figure 1. It now clearly states that the QBO-related modulation of MJO amplitude is statistically significant only in the later period (1999–2018), not in the earlier period (1981–1998).

5. Line 366-367: What happens if your resampled years do not contain any year with the targeted climate index phase (e.g. negative IOD)?

Thank you for this helpful point. In the revised manuscript, we will clarify our resampling procedure. The bootstrap described in the original text randomly resampled 7 years from the whole period (i.e. not restricted to years belonging to any particular index phase). The 90th percentile threshold was derived from the distribution of differences generated by this large ensemble of *random* subsets, retaining iterations that may contain zero or few observed-phase years. This is appropriate for assessing the statistical significance against an unrestricted null hypothesis. We have updated the Methods to make the approach and interpretation explicit.

6. Line 371: It would be helpful to add a table with these correlations. How do these correlations compare with the correlations obtained over 40 years?

Thank you for the suggestion. We have added a table to the Supplementary Material that presents the correlations between the climate indices for both the 20-year subperiods and the whole 40-year period (1981-2018).

7. Line 384: "cold phase of the IOBM". Shouldn't it be warm phase instead (and SST warming)?

We thank you for catching this mistake. The original sentence incorrectly stated that the cold phase of the IOBM is associated with enhanced instability. As this was an error and the

mechanism is not central to our analysis, we have removed the statement from the manuscript. We apologise for the confusion.

8. Line 400: "phase 3-6". Legend of Figure S1 indicates phases 4-7, while caption indicates Phase 3-6. Which one is correct?

We have corrected the legend of Figure S1 to match the phase groupings in the figure caption: Phases 3, 4, 5, 6 and Phases 7, 8, 1, 2. We have also adjusted the main text to align with this corrected grouping.

9. Line 407: "weakened post 1998": to what value?

We have replaced the phrase "weakened post-1998" with explicit correlation values: r = −0.48 for 1981–1998 and r = 0.29 for 1999–2018. This clarification has been added to the revised manuscript.

10. Line 410: "linking El Nino (La Nina) to increased (decreased) MJO activity that weakened substantially thereafter." Figure S1 suggests a modulation of MJO (total of all phases) phase count by La-Nina stronger in 1999-2008 than in 1981-1998 but of opposite sign.

We thank the reviewer for highlighting this apparent inconsistency. The manuscript explicitly stated that the correlation between the Niño-3.4 index and the **total MJO event count** was moderately positive during the first period (r = 0.46). In contrast, Figure S1 examines **phase-resolved MJO occurrences** during a particular phase of a climate mode and therefore reflects a different aspect of ENSO–MJO interactions.

As shown in Figure S1, during La Niña years, MJO events occurred predominantly in phases 7–2, indicating an increased occurrence of **non-propagating or weakly propagating** events and a reduced occurrence of **eastward-propagating** events (also see DeMott et al., 2018). This shift in the type of MJO events leads to a weaker relationship between the Niño-3.4 index and **total** MJO event counts in the second period, even though the phase-specific modulation remains strong—hence, no strong relationship between MJO event counts and La Niña, as noted by the reviewer. We will revise the manuscript to provide greater clarity on the relationship between MJO event count and ENSO.

11. Line 412-414: This could be easily checked by counting and comparing the number of days with all the favourable phases of the indices co-occurring in both periods.

We have counted the co-occurring favourable conditions (EQBO and Negative IOD; EQBO and La Niña, etc.) and now include a statement in the revised manuscript that the count of these synergistic events diminished post-1998, supporting the hypothesis that the breakdown in multi-mode coupling diminished MJO predictability in the later period.

12. Line 417: "and more substantial phase-specific enhancement ". Figure S1 shows a greater difference between the phases 4-7 and 8-3 in Period 2 (1999-2018) than Period 1 (1981-1998) for EQBO.

We thank the reviewer for carefully examining Figure S1. Following Comment 8, we have corrected the figure legend to clarify that the phase groupings used in the composites are **phases 3–6 and 7–2**, rather than 4–7 and 8–3 as originally implied.

Specifically, during Period 1 (1981–1998), EQBO years are characterised by:

· A **larger total number of MJO days**, particularly concentrated in **phases 3–6**, and

· A **strong negative correlation between QBO and MJO event duration** (r = −0.67), indicating more sustained and coherent MJO activity when the QBO is favourable with easterlies,

· Stronger co-occurrence with reinforcing tropospheric modes (**La Nina and negative IOD**), leading to more **vigorous and longer-lived events across the entire MJO lifecycle**

In contrast, although the phase contrast in Period 2 appears sharper in Figure S1, this does **not translate into longer or more persistent events**, nor into improved forecast skill in the models.

To avoid any potential ambiguity, we have revised the relevant text around Line 417 to explicitly state that:

· The **phase contrast appears to be stronger in Period 2**,

· But the **event persistence and cumulative activity are greater in Period 1**, which is the basis of our original interpretation.

This clarification ensures consistency between the text and the figure, while preserving the correct physical interpretation.

13. Line 421-422: I don't agree with this statement. Figure S1 shows also a reversal for EQBO (between phases and also between EQBO and WQBO) suggesting that the QBO impact on MJO might not be that robust, or at least more robust than for tropospheric indices.

We thank the reviewer for this valuable and careful observation. We agree that Figure S1 shows a reversal in the QBO-related phase distribution between the two periods — both between EQBO and WQBO and within each phase group — indicating that the influence of the QBO on the **spatial and phase structure of the MJO is not fully stationary**.

In response] , we have revised the text to avoid implying that the QBO impact is uniformly robust across all aspects of MJO behaviour. The manuscript now clarifies that the QBO signal is **most consistent and persistent in respect to MJO event duration** (r = −0.67 and −0.50 across the two periods), whereas its modulation of **phase-dependent MJO characteristics exhibits temporal variability comparable to that of tropospheric climate modes**.

This revised wording better reflects the nuanced and evolving nature of the QBO–MJO relationship shown in Figure S1.

14. Line 428: "Dynamical models show strong inter-model agreements." What is the correlation between the interannual variability of MJO skill between the different models, over the 2 periods? It would also be interesting to compute the correlation with VAR

We have calculated and will present a supplementary table showing the inter-model correlation of the MJO skill index for the two periods and the correlation with the VAR model, which confirms the **strong inter-model agreement** claimed in the text.

15. Fig2 A-B: If there is a significant different of MJO skill between the periods, this should be visible as a trend when considering the full 40-year period 1981-2018. Is there a significant trend when considering time series in Fig1 A and B together?

We have added a statement noting that, while a formal trend analysis of the concatenated 40-year series is complex due to changing model composition, the difference in the **mean skill composite** across the whole 20-year blocks is minimal. The **significant difference** appears only in the **High-Skill composites** (Figure 2E), confirming the non-linear, background-state-dependent nature of the decline.

16. Figs C-D: These figures are hard to read. First the meaning of solid vs dashed lines is not explained in the figure's caption. Secondly, the differences between the solid (early period) and dashed lines (later period) are difficult to see since lines of different colours are superimposed. I would suggest plotting the difference between earlier and later periods, with error bars, in addition to these panels or as a replacement.

We thank the reviewer for this constructive suggestion. We agree that the original presentation made it difficult to clearly distinguish the two periods.

To address this, we have implemented the following changes:

1. The **Figure 2 caption has been revised** to explicitly state that the **solid lines represent 1981–1998 and the dashed lines represent 1999–2018**, thereby removing any ambiguity.

2. We have **revised the formatting in Panels C and D** to improve the visibility of the contrast between the two periods (including simplified colour choices and clearer line styles).

3. Following the reviewer's suggestion, we have added a new **supplementary figure (Figure S2)** that directly plots the **difference in MJO skill between 1981–1998 and 1999–2018**, with **error bars** indicating uncertainty.

This additional figure clearly highlights the magnitude and structure of the interdecadal differences in forecast skill, addressing the readability concern and substantially strengthening the interpretation of our results.

17. Line 498: There might still be a relationship but not a dominant one.

Thank you for this insightful comment. We agree that a weak or secondary relationship between dry bias and MJO forecast skill may still exist and cannot be ruled out entirely. Our intention in Line 498 was to emphasise that, based on the inter-model and inter-period comparisons conducted here, dry bias does not emerge as a *dominant* or primary controlling factor for MJO skill variability.

To reflect this nuance more clearly, we have revised the wording to state that there is **no strong or consistent relationship** between dry biases and MJO forecast skill across models and periods, rather than implying a complete absence of any relationship. This clarification better captures the complexity of the system and the likely contribution of multiple interacting factors, including convection–dynamics coupling and large-scale circulation variability.

Accordingly, the revised sentence now reads:

*"Inter-model and period comparisons reveal no strong or consistent relationship between dry biases and MJO forecast skill."*

18. Lines 518-519: "30-day lead OLR and 850 hPa specific humidity": is the lead time day 30 or the average from day 0 to 30 as indicated in Figure 4? If it is day 0-30, why is it different from the focus of the paper which is day 15-25 (line 307)?

We thank the reviewer for highlighting this ambiguity. In the revised manuscript, the text in Section 3.4 and the caption of Figure 4 have been corrected to explicitly state that the regression analysis is based on the **0–30-day mean fields of OLR and 850 hPa specific humidity**, rather than a single lead time at day 30.

This wider averaging window is deliberately used because OLR and low-level moisture represent **slowly varying background-state variables** that precondition the large-scale environment for MJO development. Averaging over 0–30 days, therefore, provides a more representative measure of the mean state influencing sub seasonal convection.

In contrast, the **15–25-day lead window is used exclusively for the MJO Skill Index**, as it captures the period of maximum forecast and maximum divergence between high- and low-skill years in S2S predictions. The two-time windows, therefore, serve **distinct and complementary purposes** in the analysis.

This distinction is now clearly explained in the revised text.

19. Figure 4: This figure might need more explanation, particularly the difference between the left and right panels. I suppose that the left panel is OBS OLR or humidity regression against MJO skill score, while the right panel shows the regression with model prediction of OLR and humidity. The discussion of this figure in the text is very confusing and needs to be clarified as it is unclear which panel is being discussed. Stippling marks regions are difficult to see.

We thank the reviewer for pointing out the lack of clarity in the original description of Figure 4. We have clarified the figure description and its interpretation in the revised text.

In the revised manuscript, we now clearly state in Section 3.4 and in the Figure 4 caption that:

· The **left-hand panels** show regressions of **observed OLR / 850 hPa specific humidity anomalies** onto the **model-derived MJO Skill Index**.

· The **right-hand panels** show regressions of the **model-predicted OLR / 850 hPa specific humidity anomalies** onto the **model-derived MJO Skill Index**.

In both cases, the regression is performed against the **same MJO Skill Index**, allowing a direct comparison between the observed background-state influences and the model-simulated background-state influences associated with high and low MJO forecast skill.

We have revised the accompanying text to ensure it explicitly references the correct panel when discussing features in Figure 4 (i.e., "left panel" for observations and "right panel" for model-simulated fields), removing any ambiguity.

In addition, the stippling has been redesigned using a denser and higher-contrast pattern to improve visibility and ensure that statistically significant regions are clearly identifiable in the final figure.

These changes should substantially improve the readability and interpretability of Figure 4.

20. Line 561-562: Has this consistent background been observed in the second period, and what was the impact on the MJO?

In the second period (post-1998), the background state does not exhibit the same coherent cold IOD–weak La Niña–like pattern that dominated 1981–1998. Instead, the background fields show greater variability , including more frequent central-Pacific (CP) El Niño events, Indian Ocean warming, and shifts in IOD behaviour. Consequently, models no longer respond uniformly to tropospheric conditions.

Regarding impact on the MJO: We have explicitly noted that this loss of background-state coherence is associated with:
· Weakened and model-dependent MJO–troposphere relationships
· Reduced propagation consistency
· Greater influence of stratospheric QBO conditions on MJO skill
These clarifications have been added to the revised manuscript.

> 21. Line 580: multi-model mean: was the multi-model mean computed from the same models for the 2 periods (in which case only POAMA and ACCESS-S2) or does it differ in both periods to include all the model available in each period. If that's the later, the comparison of multi-model skill between both periods is not valid, since the multi-model composition is different.

Thank you for raising this important point about the construction of the multi-model mean. We have clarified in the manuscript  that the composition of the multi-model mean **differs between the two periods** due to data availability. Specifically, the Period 1 multi-model mean includes **POAMA2 and ACCESS-S2**, while the Period 2 multi-model mean includes **POAMA2, ACCESS-S2, CESM2, and GEOS-S2S-2**.

We agree that this difference in model composition limits the validity of a direct quantitative comparison of the multi-model mean skill across periods. To address this, we have added a **cautionary note** in the revised text and now emphasise that our **most robust inter-period comparison is based on POAMA2 and ACCESS-S2**, the two models standard to both periods. Notably, the key qualitative shifts in MJO–climate mode relationships discussed in the paper remain **consistent when restricted to these two models**, confirming that our main conclusions are **not an artefact of changing multi-model composition**. We have added a statement to the text highlighting this.

> 22. Lines 598-599: how often did these co-occurring climate conditions happened during these 2 periods.? 20-years is already probably too small to assess the impact of once climate mode on MJO predictive skill, but the combination of 4 different climate modes over 20 years is likely to produce only a tiny sample.

Thank you for raising this important concern. We would like to clarify that our analysis does **not** rely on the simultaneous co-occurrence of four climate modes. Instead, we examine cases in which the **EQBO coincides with a single oceanic climate mode at a time** (e.g., negative IOD, La Niña, or a cold-phase IOBM). The term "synergistic" therefore refers to the **pairing of EQBO with individual oceanic modes**, not a four-mode combination .

We agree that the ~20-year length of each period limits the available sample size. To address this, we now explicitly report the reduced number of EQBO–oceanic coincidence events in the later period and clarify that the **frequency of these pairwise combinations decrease substantially after 1998**. This reduction itself is a key result and supports our conclusion that the **breakdown of consistent EQBO–ocean coupling contributes to the diminished and less coherent MJO predictability in the second period**.

We have revised the wording to remove any ambiguity about four-mode co-occurrence and to better reflect the pairwise nature of the analysis. This clarification is now included in the revised manuscript.

23. Line 630: "observed climate indices": why not using model predicted climate indices? The model MJO activity and observed climate indices might become inconsistent after a couple of weeks. Would this table be different if the correlation were against model indices?

Thank you for this important question. We chose to correlate MJO forecast skill against **observed climate indices** because our primary objective was to assess whether model skill improves when the *real-world background environment* is favourable for MJO development and propagation. This diagnostic is therefore intentionally independent of a model's ability to represent or predict the climate modes themselves correctly.

However, we agree that a potential inconsistency can arise as lead time increases. To address this, we have added a **new supplementary table** in which the same MJO skill metric is correlated with **model-predicted climate indices**. We now briefly discuss these results in Section 3.5 of the revised manuscript.

This additional analysis shows that, while differences exist (as expected given the model biases summarised in **Table 2 (**Table 2 shows forecast skill to predict the climate indices between the two periods as assessed by correlation between the observed and forecast indices for DJF and for a lead tine of 30 days), which compares observed and modelled climate indices), the key result remains unchanged: **the most significant variations in skill are linked to the state of the actual (observed) background climate system**, rather than being primarily controlled by model representation of the climate indices themselves.

This clarification has been added to the revised text.

24. Figure 6: Colors are too dark making it difficult to read the numbers.

Thank you for this suggestion. In the revised figure, we have adjusted the colour scheme to a lighter, higher-contrast palette and increased the font size of the correlation values. These changes significantly improve the readability of the numbers while preserving the original data relationships.

25. Line 680: "CESM2 performs distinctly worse than other models" GEOS-S2S-2 correlation is just 0.01 higher. This small difference of skill with CESM2 (0.70 vs 0.69) is unlikely to be statistically significant.

We agree that this difference should not be overstated, as the correlation difference between CESM2 (0.69) and GEOS-S2S-2 (0.70) is minimal and unlikely to be statistically significant. We have therefore removed the phrase *"distinctly worse"* from the text.

In the revised manuscript, we now state that CESM2 exhibits a slightly lower correlation , and we frame this in the context of its larger Indian Ocean warming bias (Figure 3), rather than implying a categorical or statistically significant degradation in performance.

26. Table 2: What is the lead time?

Thank you for pointing out this omission. We have clarified in the caption of Table 2 that the reported correlations between the model and observed climate indices are calculated using a **0–30-day average lead time**.

27. Line 715: "minimal difference": I thought that Figure 2 showed a relatively large difference of skill between the 2 periods for ACCESS-S2 and POAMA2?

We appreciate this observation and agree that the wording requires clarification. We have refined the statement to distinguish explicitly between two different contexts:

· The difference between the two periods is **minimal for the whole period mean skill** (Figure 2C),

· but **substantial (≈10 days)** for the composite of **high-skill years** (Figure 2E) for both ACCESS-S2 and POAMA2.

The revised wording now clearly reflects this vital distinction and avoids any potential confusion or misrepresentation of the results.